# Scheduled Naps Improve Drowsiness and Quality of Nursing Care among 12-Hour Shift Nurses

**DOI:** 10.3390/ijerph18030891

**Published:** 2021-01-20

**Authors:** Kihye Han, Heejeong Hwang, Eunyoung Lim, Mirang Jung, Jihye Lee, Eunyoung Lim, Sunhee Lee, Yeon-Hee Kim, Smi Choi-Kwon, Hyang Baek

**Affiliations:** 1College of Nursing, Chung-Ang University, Seoul 06974, Korea; hankihye@cau.ac.kr; 2Department of Nursing, Asan Medical Center, Seoul 05505, Korea; lim-eunyoung@hanmail.net (E.L.); mayniang@hanmail.net (M.J.); ljihye@uiowa.edu (J.L.); eunyoung.lim.kim@gmail.com (E.L.); gemfsunhee@gmail.com (S.L.); 3Department of Clinical Nursing, University of Ulsan, Seoul 05505, Korea; kimyhee@amc.seoul.kr; 4The Research Institute of Nursing Science, College of Nursing, Seoul National University, Seoul 03080, Korea; smi@snu.ac.kr; 5School of Nursing, University of Maryland Baltimore, Baltimore, MD 21201, USA; hbaek@umaryland.edu

**Keywords:** scheduled naps, extended work hours, nurse, fatigue, drowsiness, Korea

## Abstract

Scheduled naps in the workplace are an effective countermeasure to drowsiness in safety-sensitive industries. This quasi-experimental study with a one-group, pre- and post-test design aimed to examine the effects of scheduled naps on nurses working 12-h shifts. Nurses in two pediatric intensive care units at a tertiary hospital were provided 30-min scheduled nap opportunities during their shifts. A total of 38 nurses completed pre- and post-test work diaries for sleepiness, fatigue, work demands and pace, and quality of nursing care at the end of each shift. The drowsiness of 13 nurses was continuously assessed during their shifts using infrared reflectance oculography. Nurses who reached naps reported improved levels of fatigue on the first night shift and better quality of nursing care the second night and day shifts post-test, while nurses who did not reach naps showed no significant improvements. The oculography successfully assessed drowsiness during 73% and 61% of the pre- and post-test total work hours, respectively. The total cautionary and cautionary or higher levels of drowsiness decreased. Nurse managers should consider scheduled naps in clinical settings to improve nurses’ alertness during their shifts.

## 1. Introduction

To provide patient care around the clock, hospital nurses often have nonstandard work schedules, including shift rotation and extended work hours. In Korea, many hospital nurses work traditionally in a rotating 8-h shift system [1]. Due to a nurse shortage and nurses’ need for more days off, the 12-h shift system has been recently adopted at hospitals in Korea. A Korean study reported no significant changes in patient safety incidents or prolonged overtime work hours before and after the 12-h shift system [2]. Furthermore, compared to the 8-h shift system, the 12-h shift system requires a smaller number of working days for nurses and provides greater continuity of nursing care. Nurses were more satisfied with the 12-h system but reported that their levels of fatigue accumulated and concentration sharply decreased in the last 3–4 h of their shifts.

Robust research indicates that extended work hours result in increased fatigue and drowsiness among nurses. Individuals working long work hours combined with rotating shift schedules experience more frequent sleep problems, which have detrimental effects on workers’ cognitive functioning and are well-known risk factors for decreased worker and patient safety [3]. Currently, nurses prefer the 12-h shift system, but there is a pressing need to implement effective countermeasures to fatigue and declined cognitive functioning to improve nursing quality and patient safety.

One potential strategy to reduce workplace fatigue and drowsiness of nurses working 12-h shifts is to provide a scheduled nap opportunity [4]. Since the 1970s, when sleep scientists proposed napping to reduce sleepiness during shifts, safety-sensitive industries have used napping opportunities to enhance work safety and productivity [5,6]. Many leadership organizations, including the Institute of Medicine, the Joint Commission, and the American Nurses Association, have suggested using naps to counteract the severe consequences of fatigue and sleepiness in healthcare settings [7,8,9]. Although several experimental studies have recently been conducted on hospital nurses in some Western countries [10,11], napping has not been studied in Korea. In this study, we provided scheduled naps to 12-h shift nurses based on organizational support and the scientific evidence of napping. Nap breaks were scheduled for 20–30 min between 3:00–5:00 a.m. or 3:00–5:00 p.m., which is the sleepiest time in the 24-h circadian rhythm. These naps were intended to relieve fatigue and improve arousal, thereby improving work efficiency during subsequent working hours [12,13,14].

To prevent drowsiness-related injuries and accidents, it is essential to accurately assess drowsiness. Although previous research has successfully used subjective measures of drowsiness and sleepiness [10,11], self-reported data tends to underestimate compared to objective data [15]. Physiological indicators of drowsiness and sleepiness, such as blink duration and the percentage of eyelid closure over the pupil over time, reflect the onset of sleep or micro-sleep. These indicators might not be self-reported [16]. Few studies have assessed drowsiness objectively and continuously in shift-work nurses in a natural setting during work hours.

The goals of this study were to conduct a scheduled nap intervention as a countermeasure of nurse drowsiness/fatigue in the middle of 12-h (day and night) shifts for intensive care unit (ICU) nurses and to examine whether there was an improvement in drowsiness and nursing care quality. We utilized both subjective and objective measures of drowsiness.

## 2. Materials and Methods

### 2.1. Study Design and Setting

This was a quasi-experimental study with a one-group pre- and post-test design. It was conducted in two pediatric ICUs at a tertiary hospital in Seoul, South Korea. Site one was a surgical ICU that primarily cares for children with congenital heart anomalies and those who have undergone heart surgery. Site two treats children with medical problems (e.g., cancer or congenital diseases). The two units were selected because they have implemented 12-h shift schedules since 2012, unlike the traditional 8-h work schedule of most Korean hospitals.

All participants provided written informed consent. The study protocol was approved by the Institutional Review Board of the study hospital (IRB number No. S2017-0696).

### 2.2. Recruitment/Sampling of Participants

Inclusion criteria were ICU nurses who provided direct patient care, had at least one year of experience as a registered nurse, and were working 12-h shifts, including nights (day shift: 7:00 a.m.–7:30 p.m.; night shift: 7:00 p.m.–7:30 a.m.). A total of 45 nurses (22 from site one, 23 from site two) voluntarily participated in the study at baseline. After seven nurses withdrew from the study (three transferred to another unit, two left the hospital, one went on maternity leave, and one had a schedule change to the day shift), 38 nurses completed the post-test measures (18 from site one, 20 from site two). Among the participants, 13 nurses voluntarily agreed at baseline to wear infrared reflectance (IR) oculography (Optalert, Melbourne, Australia) [16]. None of the 13 nurses dropped out during the study period. We included data from the diary of the 304 workdays and the 104 days of oculography.

### 2.3. Procedure

Participants were asked to maintain normal sleep patterns and work behaviors during the study period, without restrictions on caffeine consumption or other sleep-related behaviors.

The pretest measures were conducted from August–October of 2017. The study nurses were provided with a six-day work schedule of night-night-off-off-day-day at least once during the data collection period. All participants completed work diaries during their work schedule. The 13 nurses who volunteered to wear the oculography were monitored for their real-time drowsiness when working the two night- and two day-shifts of the six-day schedule.

The scheduled napping began in November of 2017. The principal investigator met with all of the participants and each nurse manager and provided information about the risks of nurse drowsiness and sleepiness, the scientific basis for napping, and general sleep improvement strategies. This 30-min sleep education session was designed to help nurses recognize the importance of sleeping and resting, to correct personal sleep hygiene behaviors and habits, and to maximize intervention effectiveness [17]. Nurse managers organized 30-min nap times (between 3:00 and 5:00 p.m. for day shifts and between 3:00 and 5:00 a.m. for night shifts) for each nurse at the start of their shift. When sufficient staff was present, each nurse could take a 30-min nap break in a dark, quiet, private room nearby but separate from the work units. Each room had massage chairs for use. Nurses returned to work after the 30-min nap breaks without additional time to recover full arousal. Sleep inertia likely occurs when someone is awakened during slow-wave (deep) sleep, which starts after about 30–45 min of sleep time [10].

The post-test measures were conducted in January–March of 2018. All participants completed work diaries for the two night- and two day- shifts of the six-day work schedule (i.e., night-night-off-off-day-day), as did the 13 volunteers who wore the oculography. At the end of the post-test data collection, nurse managers and staff nurses were interviewed about their napping experiences.

### 2.4. Measures

A total of 38 nurses completed the pre- and post-test work diaries. At the end of each shift, nurses were asked to report their levels of severe sleepiness and fatigue, physical and psychological work demands, work pace, and quality of nursing care provided during the shift. Each was measured using a range of scores from 1 (never) to 10 (extremely): “Please rate the highest levels of sleepiness/fatigue/physical work demands/psychological work demands/work pace/quality of nursing care provided that you perceived during the shift, respectively.” Single-item measures, often used in sleep and work diaries to reduce respondent burden, have an acceptable measurement validity [18,19,20,21]. Additionally, the participants reported the start time, end time, and location of their rest, and if napping occurred during each rest period.

Using IR oculography, the 13 nurses’ drowsiness levels were continuously assessed during work. Oculography monitors eye and eyelid movements, which are controlled by the central nervous system. When people get drowsy, the central nervous system inhibits muscles and increases the velocity and duration of blinks and other eyelid closures. The participants wore a special pair of eyeglasses (Optalert, Melbourne, Australia)) frames with IR transducers attached in the arm and positioned below and in front of the eye. These measured the duration of the separate components of eye blinks and other eye and eyelid movements. An oculography system can unobtrusively and continuously measure real-time drowsiness without electrodes or wires to infuse power. Johns Drowsiness Scale (JDS) scores were calculated and provided based on several variables, including the relative velocity and duration of blinks and other eyelid closures. JDS scores range from 0 (very alert) to 10 (very drowsy), with a score of 4.5–4.9 indicating a cautionary level of drowsiness and a score of 5.0 or above indicating a critical level of drowsiness. Increased JDS scores are highly predictive of an increased risk of severe lane excursions on a driving simulator and hazardous driving events under real-world driving conditions [16,22].

Additionally, participant characteristics were assessed at the very end of the pretest work diary, including age, marital status, having children, religious affiliation, educational attainment, number of years of registered nursing (RN) experience, and the number of years working at the current unit.

### 2.5. Data Analysis

Data were managed and analyzed using Microsoft Excel 2016 (Microsoft, Redmond, WA, USA) and SPSS Statistics for Windows Version 25.0 (Armonk, NY, USA: IBM Corp.). Descriptive analyses provided frequencies and percentages for categorical variables and means and standard deviations for continuous variables. Changes in self-reported sleepiness, fatigue, work demands, work pace, and quality of nursing care between pre- and post-testing were examined using paired t-tests. We stratified analyses by whether or not naps were reached to examine the effects of napping. We conducted separate statistical analyses for each outcome for each shift with the paired manner of the within-subject design. The time (in minutes) of when the Johns Drowsiness Scale scores were 4.5–4.9 (i.e., cautionary warning) and equal to or above 5.0 (i.e., critical warning) and 4.5 (i.e., cautionary or higher warning) at work were calculated and compared pre- and post-test using the Wilcoxon Signed-Rank Test.

## 3. Results

Table 1 presents the demographic characteristics of the study participants. Nurses were an average of 29 years old (SD = 5.0, range 23–41). More than two-thirds of the nurses were single (74%) and had no children (79%). The mean length of RN experience was 5.6 years (SD = 4.8, range 1–18), with 4.9 years at the current unit (SD = 3.6, range 1–13). Nurses who wore the drowsiness measuring device were younger than the other nurses, but otherwise were similar.

Approximately half of the nurses reached naps during their shifts at the post-test. This includes 17 and 19 nurses on their first- and second-night shifts and 16 and 18 on their first- and second-day shifts, respectively.

When comparing differences in self-reported sleepiness, fatigue, physical and psychological job demands, work pace, and quality of nursing care, all significant beneficial findings were in those who reached naps (Table 2). For example, nurses who took naps reported improved levels of fatigue on the first night shift (7.18 at pretest and 6.18 at post-test; t = 2.29, *p* = 0.04) and better quality of nursing care on the second night shift (6.79 at pretest and 7.47 at post-test; t = −2.69, *p* = 0.02) and on the second day shift (6.94 at pretest and 7.94 at post-test; t = −2.55, *p* = 0.02).

For the 13 nurses whose drowsiness levels were continuously assessed during work, the Optalert systems successfully measured their ocular movements during 73% and 61% of the total work hours at the pre- and post-tests, respectively (Table 3). The total cautionary level of drowsiness decreased from 231 to 53 min (Z = −2.07, *p* = 0.04) and the total cautionary or higher level of drowsiness decreased from 297 to 76 min (Z = −2.07, *p* = 0.04).

## 4. Discussion

This study reports a nap intervention for 12-h shift nurses in Korea. We performed an innovative, objective assessment of the real-time drowsiness of nurses during their shifts and subjective measures of work on consecutive study days at both the pre- and post-tests. According to the ocular measures, cautionary or higher levels of drowsiness decreased after the nap intervention. Furthermore, we found significant improvement in work outcomes, e.g., fatigue and quality of nursing care, after the nap intervention mostly on the second night shift. When stratifying the analyses by whether or not naps were reached, significant beneficial effects were found among those who reached naps. Our findings suggest that napping could be adopted as an effective countermeasure to drowsiness in nursing units, where napping has been limitedly implemented.

Extended work hours lead to increased fatigue and reduced alertness, which are all well-known risk factors for patient errors and occupational injuries and accidents [3]. Nurses are at high risk of sleepiness due to circadian mismatch and biological consequences, especially in combination with night shift work [23]. Nurses working in 12-h night shifts experience a substantial decline in cognitive function when working sequential nights [24]. As we found that the nap intervention was effective for the second night shift, our trial was meaningful because it indicates that even a short nap could help nurses recover from fatigue and provide better quality nursing when there is the greatest risk.

On-duty nap opportunities are reportedly effective for shift-workers and improve subjective sleepiness, fatigue, lapses, and reaction time post-nap through the end of the shift [25,26]. However, due to the wide variability in occupational conditions and job characteristics, the optimal nap setting remains undetermined. Although the duration of naps in previous research varied from 15 min to 2 h, a short nap sleep (20–30 min) might be practical in real-world occupational settings [27]. This type of sleep is usually stage 1 or 2 (light) sleep and reduces the homeostatic drive to sleep without inducing sleep inertia [28]. Both long (>1.5 h) and short (<30 min) naps reportedly help recover fatigue and increase performance [29]. The 30–90-min naps would require additional time to recover full arousal post-nap [10]. Similarly, the effects of nap timing have been mixed [25]. While much of the research in healthcare settings involves nap opportunities during shifts, both prophylactic (before shift) and mid-shift naps improve alertness and performance during the night shift [29]. As a workplace fatigue-management intervention, naps at the point of highest sleep pressure or the most tiring times during shifts would optimize the benefits of napping. When developing a scheduled nap protocol, the feasibility of scheduling work activities and break guidelines should be considered [10,26]. Furthermore, future nap interventions should include individual-level characteristics when scheduling nap breaks, such as the circadian preferences of the nurses, for the most effective and efficient nap opportunities.

Oculography successfully measured the 12-h shift nurses’ real-time drowsiness during their shifts. As its small transducer is attached to glasses frames and it continuously monitors nurses’ eye and eyelid movements, the system minimally interferes with nurses’ work. However, while its wireless system allows nurses to move around without limits, its battery sometimes runs out before the end of the shift and requires an auxiliary power bank. Nonetheless, the oculography provides objective drowsiness data, which are presented as the JDS scale. The JDS is based on a combination of oculometric variables, including the relative velocity of eyelid movements during blinking [16]. It does not require individual adjustments as it does not include subject-specific variables, such as the frequency of blinks per minute. Although oculography’s validity and acceptability have not been investigated in Korea, our findings support its applicability in safety-sensitive healthcare settings.

Our study findings should be interpreted with caution. First, the relatively small sample of nurses (*n* = 38) limits the generalizability of our results. However, the study nurses provided self-reported data at each pre- and post-test during a total of 304 shifts and objective data on the level of drowsiness during 104 shifts. Furthermore, this one group pre- and post-test design did not allow for control group comparison. Participant bias might affect the results, especially given that the sleep education session was provided after pretest measures were collected. Nevertheless, repeated measures during multiple days of work at the pre- and post-tests partially support study validity. Additionally, we did not include some potential factors that might affect drowsiness at work, such as history, comorbidities (e.g., depression, anxiety, obesity, or sleep apnea), health behaviors (e.g., smoking, drinking alcohol, and exercising), and characteristics related to their commute. Lastly, due to the five-month gap between the pre- and post-tests, seasonal variation might have affected the study results. Nurses may have been less drowsy at the post-test because they were busier.

## 5. Conclusions

Nurses with extended work hours are often drowsy while working, leading to increased risk to their and their patient’s safety. Since the 1970s, napping has been implemented as an effective countermeasure to fatigue and sleepiness in many safety-sensitive industries, such as aviation, transportation, and manufacturing [6]. However, napping in healthcare settings, in which many professionals work around-the-clock, has not been frequently adopted, especially in Korea. Our study implemented napping at two ICU settings and found that the practice was well-accepted by and helpful to nurses. To mitigate the risk of fatigue and drowsiness, nurse managers should consider scheduled nap strategies in clinical settings. Contextual barriers should be thoroughly investigated to promote the applicability and feasibility of napping in nursing settings. While staffing inadequacies and a fast work pace were reported in a U.S. study as potential barriers against napping in nursing units, work environments in Korea have not been evaluated.

Future research on napping in a healthcare setting should include a larger sample size with a variety of work settings. Follow-up studies would be designed with a larger sample size and a control group to increase the reliability of the results. Further use of the oculography should be applied in nursing research to better understand healthcare workers’ drowsiness and work performance in safety-sensitive work settings.

## Figures and Tables

**Table 1 ijerph-18-00891-t001:** Participant pretest characteristics (*n* = 38).

Characteristics	Categories	Total (*n* = 38)		Those Wearing the Drowsiness Measuring Device (*n* = 13)	
		Frequency	Mean	Frequency	Mean
Age			29.1 (SD = 5.0, range 23–41)		27.5 (SD = 3.2, range 24–35)
Marital status	Single	28 (73.7%)		11 (84.6%)	
	Married	10 (26.3%)		2 (15.4%)	
Having children	None	30 (78.9%)		11 (84.6%)	
	1 child	4 (10.5%)		0 (0.0%)	
	2 or more children	4 (10.5%)		2 (15.4%)	
Religion	Yes	22 (57.9%)		9 (69.2%)	
	No	16 (42.1%)		4 (30.8%)	
Educational attainment	Associate’s	1 (2.6%)		1 (7.7%)	
	BSN	34 (89.5%)		11 (84.6%)	
	Master’s or higher	3 (7.9%)		1 (7.7%)	
Years of RN experience			5.6 (SD = 4.8, range 1–17.7)		4.4 (SD = 3.7, range 1.0–13.0)
Years of working at the current unit			4.9 (SD = 3.6, range 1–12.8)		4.2 (SD = 3.3, range 1.0–10.75)

**Table 2 ijerph-18-00891-t002:** Differences in self-reported sleepiness, fatigue, physical and psychological job demands, work pace, and quality of nursing care between pre- and post- tests, stratified by whether naps were reached.

Variables	Individuals Who Reached Naps								Individuals Who Did Not Reach Naps							
	*n*	Pretest		Post-Test		Paired Differences		*p*	*n*	Pretest		Post-Test		Paired Differences		*p*
		Mean	SD	Mean	SD	Mean	SD			Mean	SD	Mean	SD	Mean	SD	
**NIGHT 1**																
Sleepiness	17	6.29	1.72	5.47	1.84	0.82	2.04	0.12	21	5.70	2.23	5.85	2.28	−0.15	2.92	0.82
Fatigue		7.18	1.47	6.18	1.59	1.00	1.80	0.04		7.15	1.39	6.70	1.95	0.45	2.50	0.43
Physical demands		6.82	1.74	6.29	1.76	0.53	2.21	0.34		7.30	1.75	7.25	1.74	0.05	2.11	0.92
Psychological demands		6.82	1.47	6.53	1.81	0.29	1.86	0.52		7.10	1.92	7.45	1.99	−0.35	2.11	0.47
Work pace, fast		5.71	2.11	5.41	2.24	0.29	2.44	0.63		6.50	2.14	6.65	2.46	−0.15	2.78	0.81
Quality of nursing care		7.18	1.19	7.65	1.32	−0.47	1.18	0.12		7.80	1.20	7.40	1.23	0.40	1.35	0.20
**NIGHT 2**																
Sleepiness	19	6.63	1.92	6.32	2.19	0.32	2.14	0.53	19	7.21	1.72	6.53	2.22	0.68	2.26	0.20
Fatigue		8.05	1.54	7.16	1.80	0.89	2.23	0.10		8.26	1.19	7.58	1.57	0.68	1.70	0.10
Physical demands		7.58	1.54	6.21	1.69	1.37	2.50	0.03		7.11	1.94	7.26	1.45	−0.16	2.06	0.74
Psychological demands		7.68	1.38	6.42	1.77	1.26	2.21	0.02		7.05	1.96	7.47	1.39	−0.42	1.77	0.32
Work pace, fast		6.84	2.24	5.37	2.17	1.47	2.37	0.01		6.32	1.89	6.53	1.74	−0.21	1.78	0.61
Quality of nursing care		6.79	1.51	7.47	1.43	−0.68	1.11	0.02		7.05	1.72	7.47	1.07	−0.42	1.12	0.12
**DAY 1**																
Sleepiness	16	4.38	2.58	5.38	2.55	−1.00	3.18	0.23	22	5.09	2.49	4.41	2.34	0.68	3.00	0.30
Fatigue		6.31	2.27	6.88	1.86	−0.56	2.83	0.44		7.00	1.98	6.73	2.19	0.27	2.45	0.61
Physical demands		7.00	1.37	6.50	1.75	0.50	1.97	0.33		7.23	1.97	7.68	1.86	−0.45	1.79	0.25
Psychological demands		7.00	1.46	6.81	1.64	0.19	1.38	0.59		7.41	2.04	7.68	1.99	−0.27	1.61	0.44
Work pace, fast		6.44	2.00	6.25	2.11	0.19	1.87	0.69		7.00	2.37	7.50	2.48	−0.50	2.32	−1.01
Quality of nursing care		7.38	0.81	7.38	0.89	0.00	0.82	1.00		7.41	1.56	7.05	1.86	0.36	1.87	0.91
**DAY 2**																
Sleepiness	18	5.29	2.80	5.24	2.95	0.06	3.63	0.95	20	4.95	2.28	5.20	1.96	−0.25	2.63	−0.43
Fatigue		7.59	2.00	6.94	2.56	0.65	2.55	0.31		6.80	1.54	7.00	2.32	−0.20	2.57	−0.35
Physical demands		7.65	1.66	7.29	1.99	0.35	2.23	0.52		6.85	1.53	7.60	1.79	−0.75	2.67	−1.26
Psychological demands		8.06	1.34	7.77	1.68	0.29	1.86	0.52		6.95	1.99	7.65	1.69	−0.70	2.60	−1.21
Work pace, fast		7.41	2.21	6.94	1.92	0.47	2.60	0.47		6.15	2.18	6.90	2.34	−0.75	2.83	−1.19
Quality of nursing care		6.94	1.71	7.94	1.03	−1.00	1.62	0.02		7.15	1.63	7.10	1.25	0.05	1.28	0.18

**Table 3 ijerph-18-00891-t003:** Drowsiness monitoring pre- and post-test (*n* = 13).

Total Time the Device Was Worn		Pretest			Post-Test			Wilcoxon Signed-Rank Test		
		459:27:30 (73.4%)			381:02:48 (61.1%)					
Level of Drowsiness		Cautionary	Critical	≥Cautionary	Cautionary	Critical	≥Cautionary	Cautionary	Critical	≥Cautionary
Time (in min)	Night 1	40	23	63	17	4	21	*p* = 0.16	*p* = 0.14	*p* = 0.12
	Night 2	66	7	73	15	4	19	*p* = 0.69	*p* = 0.99	*p* = 0.69
	Day 1	34	8	42	10	4	14	*p* = 0.14	*p* = 0.11	*p* = 0.14
	Day 2	91	28	119	11	11	22	*p* = 0.08	*p* = 0.14	*p* = 0.08
	Overall	231	66	297	53	23	76	*p* = 0.04	*p* = 0.09	*p* = 0.04

## Data Availability

Due to the nature of this research, participants of this study did not agree for their data to be shared publicly, so supporting data is not available.

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
