# Peer review of "Scheduled Naps Improve Drowsiness and Quality of Nursing Care among 12-Hour Shift Nurses"

_ijerph, 2021, doi:10.3390/ijerph18030891_

Round 1

Reviewer 1 Report

Dear Authors,

Thank you for being diligent in addressing all concerns. It is a neat study and hopefully other researchers will be able to build upon the results and further broaden the scope. I would recommend to design follow-up studies with a larger sample size derived using a priori performed power analysis and a control group to increase the reliability of the results

Kind regards

Author Response

Response to Reviewer 1 Comments

Yes        Can be improved            Must be improved          Not applicable

Does the introduction provide sufficient background and include all relevant references?

(x)         ( )          ( )         ( )

Is the research design appropriate?

( )          (x)         ( )         ( )

Are the methods adequately described?

(x)         ( )          ( )         ( )

Are the results clearly presented?

(x)         ( )          ( )         ( )

Are the conclusions supported by the results?

(x)         ( )          ( )         ( )

Comments and Suggestions for Authors

Dear Authors,

Thank you for being diligent in addressing all concerns. It is a neat study and hopefully other researchers will be able to build upon the results and further broaden the scope. I would recommend to design follow-up studies with a larger sample size derived using a priori performed power analysis and a control group to increase the reliability of the results

Kind regards

Thank you very much for your recommendation. Further recommendations for the future studies were added in the Conclusion section lines 256-257).

Reviewer 2 Report

Dear authors,

The paper was well-revised. I have some minor suggestions:

You do not need to present all results of statistical analyses. I suggest removing “t” value in table 2, and “Z” value in table 3. Or remove Z and p-value in table 3, and mark “*” when p-value is lower than 0.05 in table 3.

I could not see the p-value of “≥ Cautionary” because the table 3 was shown with track changes.

In addition, you said that you had added the discussion about nap interventions in “line 208-2017”, but the line number was wrong because of track change.

Author Response

Response to Reviewer 2 Comments

Yes        Can be improved            Must be improved          Not applicable

Does the introduction provide sufficient background and include all relevant references?

(x)         ( )          ( )         ( )

Is the research design appropriate?

(x)         ( )          ( )         ( )

Are the methods adequately described?

(x)         ( )          ( )         ( )

Are the results clearly presented?

( )          (x)         ( )         ( )

Are the conclusions supported by the results?

(x)         ( )          ( )         ( )

Comments and Suggestions for Authors

Dear authors,

The paper was well-revised. I have some minor suggestions:

  1. You do not need to present all results of statistical analyses. I suggest removing “t” value in table 2, and “Z” value in table 3. Or remove Z and p-value in table 3, and mark “*” when p-value is lower than 0.05 in table 3.

Thank you for your suggestions. We removed t values from Table 2 and Z values from Table 3 as suggested.

  1. I could not see the p-value of “≥ Cautionary” because the table 3 was shown with track changes.

We revised the table format to show the all statistical values (Table 3).

  1. In addition, you said that you had added the discussion about nap interventions in “line 208-2017”, but the line number was wrong because of track change.

We revised the Discussion section format to show the all contents (lines 207-216).

This manuscript is a resubmission of an earlier submission. The following is a list of the peer review reports and author responses from that submission.

Round 1

Reviewer 1 Report

The authors investigated whether scheduled naps (30-min) during the period of most sleepiness of a 12-h work shift could affect self-reported sleepiness, fatigue, quality of nursing care, and objectively assessed drowsiness in nurses. Results showed that self-reported fatigue and quality of nursing care were significantly improved for a sequential night shift, following intervention naps. A marked reduction in drowsiness was also evident following intervention naps.

Three primary strengths of the study are – (1) Involving a shift-working demographic (nurses in Korea) not studied previously, (2) Within-subjects design, and (3) Objectively assessing drowsiness. Lack of control group is a major limitation of this otherwise well-drafted study. Few comments below could further improve the readability of the manuscript –

Please mention if any statistical corrective measure was employed to account for multiple comparisons in analyses.

It has been reported that about half the nurses “reached naps” post-test. Do the authors have data on nap durations reached? If yes, perhaps it can be used as a covariate in the analyses. Also, does including only the nurses that “reached naps” change analyses outcomes?

It will be useful if authors discuss what additional measures can be taken to increases the effectiveness of the nap interventions. Can the nap duration be optimized? Can the time of nap be optimized based upon circadian preference of the nurse (e.g. chronotype)? Please review - https://www.tandfonline.com/doi/full/10.1080/10903127.2017.1376136

Authors need to acknowledge a “participant bias” in the subjective measures, especially given the “sleep education session” was planned after pre-test measures were collected.

Author Response

Response to Reviewer 1 Comments

The authors investigated whether scheduled naps (30-min) during the period of most sleepiness of a 12-h work shift could affect self-reported sleepiness, fatigue, quality of nursing care, and objectively assessed drowsiness in nurses. Results showed that self-reported fatigue and quality of nursing care were significantly improved for a sequential night shift, following intervention naps. A marked reduction in drowsiness was also evident following intervention naps.

Three primary strengths of the study are – (1) Involving a shift-working demographic (nurses in Korea) not studied previously, (2) Within-subjects design, and (3) Objectively assessing drowsiness. Lack of control group is a major limitation of this otherwise well-drafted study. Few comments below could further improve the readability of the manuscript

We greatly appreciate your thoughtful comments, which helped to improve our manuscript.

  1. Please mention if any statistical corrective measure was employed to account for multiple comparisons in analyses.

All participants worked the same work schedules (i.e., night-night-off-off-day-day) and provided data on each shift. We conducted separate statistical analyses for each outcome per shift with the paired manner of the within-subject design. Due to the small sample size (n = 38), we could not include corrective measures to account for multiple comparisons in the analyses. We have clarified this in the Analysis section (Page 4, lines 150-153).

  1. It has been reported that about half the nurses “reached naps” post-test. Do the authors have data on nap durations reached? If yes, perhaps it can be used as a covariate in the analyses. Also, does including only the nurses that “reached naps” change analyses outcomes?

To account for whether or not the nurses reached naps, we stratified the analysis by the factor and found significant beneficial results only among those who reached naps post-test (Page 4, lines 149-150 in the Data Analysis section; Page 5, lines 176-181 in the Results section and the Supplementary File; and Page 6, lines 198-199 in the Discussion section).

  1. It will be useful if authors discuss what additional measures can be taken to increases the effectiveness of the nap interventions. Can the nap duration be optimized? Can the time of nap be optimized based upon circadian preference of the nurse (e.g. chronotype)? Please review - https://www.tandfonline.com/doi/full/10.1080/10903127.2017.1376136

Thank you for your recommendation. We expanded the discussion regarding potential strategies to increase the effectiveness of naps, including the optimal duration and timing of naps (Page 6, lines 210-219).

  1. Authors need to acknowledge a “participant bias” in the subjective measures, especially given the “sleep education session” was planned after pre-test measures were collected.

Thank you very much for your suggestion. We have added participant bias in the limitation section (Page 6, lines 234-235).

 (Page 3, line 142).

Reviewer 2 Report

Dear authors,

This research was intervention study to evaluate the effect of napping on drowsiness and nursing care quality in 12-hour shift nurses. The paper in interesting, and my recommendations are following:

  1. We can wake up easily during shallow sleep (stages 1-2) but waking up during deep sleep (stages 3-4) is difficult, which takes time to be fully awakened. In general, the time to reach deep sleep phase is about 20 minutes, so 20 minutes of nap is recommended to wake up quickly. In addition, total nap time should be provided about 40 minutes, considering the time to fall asleep (10 minutes) and the time to fully awaken after nap (10 minutes). Therefore, it is desirable to wake up the workers 10 minutes before the end of nap to give them some time for recover arousal.

In your research, you planned 30 minutes of nap. Did you provide the time to recover full arousal after nap, or did the nurses return to work immediately after nap?

  1. You described half of the nurses reached naps at the post-test. I think whether the nurses reached naps may have influenced a lot on the outcome. But the result did not show this. I recommend you should perform the analyses controlling this factor (reaching nap), or stratified analyses with this factor.

  1. I think the statistics of the table 3 is not derived from statistical analysis. If I am right, why didn’t you perform statistical analysis for JDS scores (cautionary and critical time in minutes)?

  1. The discussion is not sufficient. I think there have been many researches on the effect of nap in the workers. Please review previous studies and add the description for discussion of them.

  1. Many factors that may affect drowsiness on duty were not included in the study: past history and comorbidity (especially depression, anxiety, sleep apnea, etc.), body mass index, social history (smoking, alcohol drinking, exercise), commute method and commuting time, and so on. If you cannot include theses factors in the paper, you should describe this as a limitation.

  1. Please use full-term for RN (line 153).

Author Response

Response to Reviewer 2 Comments

This research was intervention study to evaluate the effect of napping on drowsiness and nursing care quality in 12-hour shift nurses. The paper in interesting, and my recommendations are following:

We greatly appreciate your thoughtful comments, which have improved the manuscript.

  1. We can wake up easily during shallow sleep (stages 1-2) but waking up during deep sleep (stages 3-4) is difficult, which takes time to be fully awakened. In general, the time to reach deep sleep phase is about 20 minutes, so 20 minutes of nap is recommended to wake up quickly. In addition, total nap time should be provided about 40 minutes, considering the time to fall asleep (10 minutes) and the time to fully awaken after nap (10 minutes). Therefore, it is desirable to wake up the workers 10 minutes before the end of nap to give them some time for recover arousal.

In your research, you planned 30 minutes of nap. Did you provide the time to recover full arousal after nap, or did the nurses return to work immediately after nap?

Our nurse participants should return to work after the 30-minute nap breaks without additional time to recover full arousal post-nap because the risk for sleep inertia reportedly increases for naps longer than 30 minutes (Geiger-Brown et al., 2016). We have clarified this in the Procedure section (Page 3, lines 109-111).

  1. You described half of the nurses reached naps at the post-test. I think whether the nurses reached naps may have influenced a lot on the outcome. But the result did not show this. I recommend you should perform the analyses controlling this factor (reaching nap), or stratified analyses with this factor.

Thank you very much for your suggestion. We have stratified the analysis by whether or not naps were reached and found significant beneficial results among those who reached naps post-test (Page 4, lines 149-150 in the Data Analysis section; Page 5, lines 176-181 in the Results section and the Supplementary File; and Page 6, lines 198-199 in the Discussion section).

  1. I think the statistics of the table 3 is not derived from statistical analysis. If I am right, why didn’t you perform statistical analysis for JDS scores (cautionary and critical time in minutes)?

We could not perform inferential statistics for the JDS scores due to the small sample size (n = 13). We additionally conducted the generalized mixed model with a Poisson link and found significant reductions in cautionary and critical times in minutes post-test. We did not present the results due to the possibility of overfitting. We mentioned this in the Results section (Page 6, lines 186-189). 

  1. The discussion is not sufficient. I think there have been many researches on the effect of nap in the workers. Please review previous studies and add the description for discussion of them.

Thank you for your recommendation. We have expanded the discussion about the effects of naps in workers (Page 6, lines 210-219).

  1. Many factors that may affect drowsiness on duty were not included in the study: past history and comorbidity (especially depression, anxiety, sleep apnea, etc.), body mass index, social history (smoking, alcohol drinking, exercise), commute method and commuting time, and so on. If you cannot include these factors in the paper, you should describe this as a limitation.

Thank you very much for pointing this out. We have mentioned potential factors that might affect drowsiness while on shift, which were included in our study in the Limitation section (Page 7, lines 237-239).

  1. Please use full-term for RN (line 153).

We wrote out “registered nurses” where “RN” is first mentioned in the manuscript (Page 3, line 142).

Round 2

Reviewer 2 Report

Dear authors,

Your study was intervention study, and I think you can improve your paper further. My recommendations are following:

  1. The title of your study is “scheduled naps improve drowsiness and quality of nursing care”, and the purpose was “to evaluate whether there was an improvement in drowsiness and nursing care quality with scheduled nap intervention”. Among 38 subjects, about a half had taken a nap, and others had not. If you would suggest the nap can improve drowsiness and quality of nursing care, the findings should be derived from the statistical analyses with those who had taken naps. For this, you should exclude the subjects who had not taken naps or perform stratified analyses and show the improvement was only found in the subjects who had taken naps. I recommend including the supplementary table in the main text (table 2 is not necessary, so you can remove it).

(If the paper incudes only table 1 – 3 and not include supplementary table, the title should be changed into “scheduled rest during night work can improve…”).

  1. To conclude the naps can improve drowsiness, the finding should be supported by the results of study. Sleepiness was not statistically significant both day and night work in the subjects who had taken naps in the supplementary table, and the statistical analyses of drowsiness monitoring was not performed. I do not think the result can support that “naps can improve drowsiness”. So, you should perform statistical analyses of drowsiness monitoring (JDS score, table 3) to evaluate whether the change was statistically significant. Of course, the number of subjects (13) was small, but you can try non-parametric test (such as Wilcoxon signed rank test).

  1. You added the discussion about the naps, but it is not sufficient.
  • Naps > 30 min may lead to sleep inertia, why?
  • There have been many studies on the nap duration and nap timing in shift workers (long nap (1.5 – 2 hours), short nap (20 – 40 mins), nap before night work, nap during night work, and so on), and effect of naps in shift workers. You should review previous studies and add some discussion.